# SparseDM: Toward Sparse Efficient Diffusion Models

## Abstract

Diffusion models represent a powerful family of generative models widely used for image and video generation. However, the time-consuming deployment, long inference time, and requirements on large memory hinder their applications on resource constrained devices. In this paper, we propose a method based on the improved Straight-Through Estimator to improve the deployment efficiency of diffusion models. Specifically, we add sparse masks to the Convolution and Linear layers in a pre-trained diffusion model, then transfer learn the sparse model during the fine-tuning stage and turn on the sparse masks during inference. Experimental results on a Transformer and UNet-based diffusion models demonstrate that our method reduces MACs by $50\%$ while increasing FID by only 0.44 on average. Sparse models are accelerated by approximately 1.2x on the GPU. Under other MACs conditions, the FID is also lower than 1 compared to other methods.

## 1 Introduction

Diffusion Models (DM) (Song et al., 2021; Karras et al., 2022) have been one of the core generative modules in various computer vision tasks. Generally, they are composed of two parts: a forward/diffusion process that perturbs the data distribution to learn the time-dependent score functions, and a reverse/sampling process that generates data samples from a prior distribution in an iterative manner. Though diffusion models have advantages on both sample quality and mode coverage over other competitors, their slow inference speed and heavy computational load during the inference process inevitably restrict their applications on most mobile devices.

To reduce the computational load in the inference process of diffusion models, various methods have been proposed to minimize the number of inference steps, such as the training-free samplers (Bao et al., 2022; Zhao et al., 2023; Lu et al., 2022; 2023; Zheng et al., 2023) and the distillation methods (Salimans & Ho, 2022; Luo et al., 2023b;a). However, their sample quality is still unsatisfactory as a few sampling steps cannot faithfully reconstruct the high-dimensional data space, e.g., the image or video samples. This problem is even more pronounced on mobile devices with limited computing capabilities. Simultaneously, a few works explore reducing the Multiple-Accumulate operations (MACs) at each inference step (Bolya & Hoffman, 2023; Fang et al., 2023b). However, their work cannot be accelerated on GPUs. Since DM is an intensive model parameter calculation, NVIDIA Ampere architecture GPU supports a 2:4 sparse (4-weights contain 2 non-zero values) model calculation, which can achieve nearly 2 times calculation acceleration (Pool & Yu, 2021; Mishra et al., 2021). Although structural pruning (Fang et al., 2023b) has been used in DM, 2:4 structured sparsity inference has not been implemented. Our method aims to reduce MACs at each step through 2:4 and other scale sparsity.

Recently, several popular computational architectures, such as NVIDIA Ampere architecture and Hopper GPUs, have developed acceleration methods for model inference and have been equipped with fine-grained structured sparse capabilities. A common requirement of these acceleration techniques is the 2:4 sparse mode, which only preserves 2 of the 4 adjacent weights of a pre-trained model, *i.e.*, requires a sparsity rate of $50\%$. Given this sparsity, the acceleration techniques only process the non-zero values in matrix multiplications, theoretically achieving a 2x speedup. In essence, a diffusion model supporting the 2:4 sparsity mode can reduce $50\%$ computational load at each inference step, which is valuable when considering the iterative refinement process in a generation. To the best of our knowledge, previous works have not explored maintaining the sample quality of dif-

Table 1: The results of manually redesigning the U-ViT model and sparse pruning the U-ViT model using ASP. UViT Small is from the U-ViT method. Half UViT Small is set to half the depth and number of heads of UViT Small. ASP is the method of sparse pruning from Nvidia.

| Datasets | Models | Methods | FID | MACs (G) |
|---|---|---|---|---|
| CIFAR10 32x32 | UViT Small | U-ViT | 3.97 | 11.34 |
| | Half UViT Small | U-ViT | 678.20 | 5.83 |
| | UViT Small | ASP | 319.87 | 5.76 |
| CelebA 64x64 | UViT Small | U-ViT | 3.30 | 11.34 |
| | Half UViT Small | U-ViT | 441.37 | 5.83 |
| | UViT Small | ASP | 438.31 | 5.76 |

fusion models with a 2:4 or other scale sparsity mode, which motivates us to present the techniques in this work.

We use a state-of-the-art Transformer-based DM, named U-ViT (Bao et al., 2023), to analyze the shortcomings of existing 2:4 structured sparse tools. Also, we manually design a half-size DM network to test FID and MACs. From Table 1, we can see that simply reducing the U-ViT model parameters by nearly 50% may cause FID to collapse catastrophically. Automatic Sparsity (ASP) (Pool & Yu, 2021; Mishra et al., 2021)) for model sparse training consumes a lot of GPU time, but FID is poor. Therefore, for DM, we need to redesign the pruning method to reduce the amount of calculation and maintain FID as much as possible. This paper proposes a new method for implementing 2:4 structured sparsity and other scale sparse inference. (1) Masks with different sparsity rates are applied to convolutional and linear layers. (2) Sparse regularization is added to back-propagation, improving the STE method to train sparse models. (3) Knowledge is gradually transferred from dense models to sparse models to improve models' performance with high sparsity rates.

Our contributions can be listed as follows:

- We propose a transfer learn sparse masks method that achieves 2:4 structured sparse inference and other scale sparse pruning for DMs with a Transformer or UNet backbones.

- We conduct experiments on four datasets. The average FID of 2:4 sparse DM is only increased by 0.44 compared with the dense model. The inference acceleration on the GPU is approximately 1.2x.

- Testing nine scales, with similar MACs for each scale, our sparse inference FID is 1 lower than other methods.

## 2 PROBLEM FORMULATION

### 2.1 DIFFUSION MODELS

The diffusion model is divided into a forward process and a backward process. The forward process is a step-by-step process of adding noise to the original image to generate a noisy image, generally formalized as a Markov chain process. The reverse process is to remove noise from a noisy image and restore the original image as much as possible. Gaussian mode was adopted to approximate the ground truth reverse transition of the Markov chain. The training process of diffusion models is the process of establishing noise prediction models. The loss function of DM is minimizing a noise prediction objective. The forward process is formalized as:

$$q(\boldsymbol{x}_{1:T}|\boldsymbol{x}_0) = \prod_{t=1}^{T} q(\boldsymbol{x}_t|\boldsymbol{x}_{t-1}), \tag{1}$$

where $\boldsymbol{x}_t$ is input data at $t$. In the backward process, the mean network of the denoising transition probability is formalized as:

$$\boldsymbol{\mu}_t^*(\boldsymbol{x}_t) = \frac{1}{\sqrt{\alpha_t}} \left( \boldsymbol{x}_t - \frac{\beta_t}{\sqrt{1-\overline{\alpha}_t}} \mathbb{E}[\boldsymbol{\epsilon}|\boldsymbol{x}_t] \right), \tag{2}$$

where $\alpha_t$ and $\beta_t$ are the noise schedule at $t$, $\alpha_t + \beta_t = 1$. $\overline{\alpha}_t = \prod_{i=1}^{t} \alpha_i$, and $\epsilon$ is the standard Gaussian noises added to $\boldsymbol{x}_t$. The DM training is a task of minimizing the noisy prediction errors, expressed as $\min_{\mathcal{W}} \mathcal{L}(\mathcal{W}; \mathcal{D})$, where $\mathcal{D}$ is dataset, $\mathcal{L}$ is loss function, $\mathcal{W}$ is dense weights. The DM inference, data generation, is expressed as $\boldsymbol{y}_t = \mathcal{F}(\mathcal{W}; \boldsymbol{x}_t)$, where $\mathcal{F}$ is trained DM, $\boldsymbol{y}_t$ is the output of DM inference.

## 2.2 SPARSE PRUNING

Sparse network computation is an effective method to reduce MACs in deep networks and accelerate computation. Currently, one-shot sparse training and progressive sparse training are commonly used.

The one-shot training method is easy to use, and the steps are as follows (Mishra et al., 2021; Pool & Yu, 2021): Train a regular dense model. Prune the weights on the fully connected and convolutional layers in a 2:4 sparse mode. Retrain the pruned model. The one-shot pruning is expressed as:

$$\widetilde{\mathcal{W}} = \mathcal{W} \odot \mathcal{M}, \tag{3}$$

where, $\widetilde{\mathcal{W}}$ is the sparse weight after the dense weight $\mathcal{W}$ is pruned, recorded as $\widetilde{\mathcal{W}} \leftarrow \text{Pruning}(\mathcal{W})$, $\mathcal{M}$ is a 0-1 mask, $\mathcal{M}$ is the $(N{:}M)$ sparse mask ($M$-weights contain $N$ non-zero values), $\odot$ represents element-wise multiplication.

One-shot sparse pruning can cover most tasks and achieve speedup without losing accuracy. However, for some challenging tasks that are sensitive to changes in weight values, doing sparse training for all weights at once will result in a large amount of information loss (Han et al., 2015). With the same number of tuning iterations, progressive sparse training can achieve higher model accuracy than one-shot sparse training. Suppose there are $k$ masks $[\mathcal{M}_1, \mathcal{M}_2, ..., \mathcal{M}_k]$, the progressive sparse process is formalized as:

$$\widetilde{\mathcal{W}} = \mathcal{W} \odot \mathcal{M}_i, \ \ \mathcal{M}_i \in [\mathcal{M}_1, \mathcal{M}_2, ..., \mathcal{M}_k], \tag{4}$$

The reason why traditional progressive sparse training methods work well is to reuse the knowledge of dense models as much as possible. However, this progressive sparse training method has only proven effective when the training data distribution is stable, such as when training a CNN classification model.

## 2.3 DISTRIBUTION SHIFT ON DM TRAINING AND SPARSE PRUNING

Robust Fairness Regularization (RFR) (Jiang et al., 2024) improved that distribution shift can be transformed as data perturbation, and data perturbation and model weight perturbation are equivalent for classifier models. Assuming that the perturbation of the label is not considered, the equivalent of data perturbation and model weight perturbation is formalized as:

$$\mathbb{E}_{\delta_{\mathcal{D}}(\mathcal{D})}\mathbb{E}_{(\mathcal{D})\sim\mathcal{P}}[\mathcal{L}(\mathcal{F}_{\mathcal{W}}(\mathcal{D} + \delta_{\mathcal{D}}(\mathcal{D}))] = \mathbb{E}_{(\mathcal{D})\sim\mathcal{P}}[\mathcal{L}(\mathcal{F}_{\mathcal{W}+\Delta\mathcal{W}}(\mathcal{D}))], \tag{5}$$

where the training dataset $\mathcal{D}$ with distribution $\mathcal{P}$, suppose the training dataset is perturbed with data perturbation $\delta$, and the neural network is given by $\mathcal{F}_{\mathcal{W}}(\cdot)$, for the general case, there exists model weight pruned as perturbation $\Delta\mathcal{W}$, so that the training loss $\mathcal{L}$ on perturbed training dataset is the same with that for model weight perturbation $\Delta\mathcal{W}$ on training distribution.

The training process of DMs generally involves only the distribution shift of the noisy data. We believe that DM training adds Gaussian noise to the data, which perturbs the original image data distribution. Existing sparse pruning methods generally do not consider the distribution shift of noisy data but only the model's weight changes. Inspired by RFR's conclusions, we convert the distribution shifts caused by changes in model weights into distribution shifts caused by data changes for the DM's training process. The DM's sparse ratio is fixed, and then the knowledge of the dense model is transferred to the sparse model.

## 3 SPARSE FINETUNING DM

In this section, we introduce our proposed framework in detail. We start with the overall framework of our proposal. Then, we present the finetuning DM with sparse masks to reduce MACs and prepare for 2:4 sparse GPU acceleration. Moreover, we introduce the training and inference with sparse masks to enhance the DM. Fig. 1 shows an overview of the proposed framework.

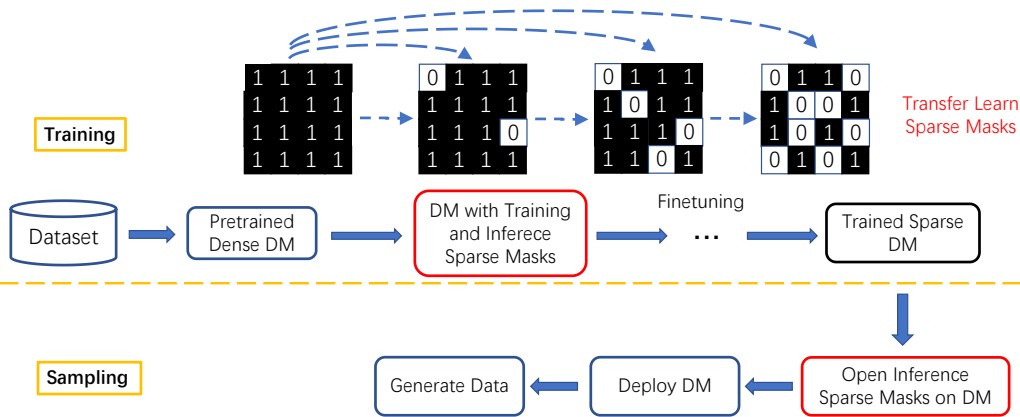

Figure 1: Framework Overview. This includes the process of transfer learning sparse models.

## 3.1 STRAIGHT-THROUGH ESTIMATOR FOR SPARSE TRAINING

A straightforward solution for training an $N{:}M$ sparsity network is to extend the Straight-through Estimator (STE) (Bengio et al., 2013) to perform online magnitude-based pruning and sparse parameter updating. The concrete formula is described as:

$$\mathcal{W}_{t+1} \leftarrow \mathcal{W}_t - \gamma_t g(\widetilde{\mathcal{W}_t^{(N:M)}}), \tag{6}$$

where $\gamma_t$ is learning rate at time $t$, and $g$ is the gradient. In STE, a dense network is maintained during the training process. In the forward pass, the dense weights $\mathcal{W}$ are projected into sparse weights $\widetilde{\mathcal{W}^{(N:M)}} = S(\mathcal{W}, N, M)$ satisfying $N{:}M$ sparsity, and here $S(\cdot)$ is a projection function. The sparse DM training task is expressed as:

$$\min_{S(\mathcal{W}, N, M)} \mathcal{L}(\mathcal{W}; \mathcal{D}), \tag{7}$$

where $\mathcal{L}$ is the loss function for training DM. The sparse DM inference is expressed as:

$$\boldsymbol{y}_t^{(N:M)} = \mathcal{F}(\widetilde{\mathcal{W}^{(N:M)}}; \boldsymbol{x}_t), \tag{8}$$

where $\boldsymbol{x}_t$ is input data at $t$, $\boldsymbol{y}_t^{(N:M)}$ is the inference output of $N{:}M$ sparse DM at time $t$. We incorporate sparse mask information into the backward propagation process to mitigate the negative impact of the approximate gradient calculated by vanilla STE. The updated formula is modified as follows:

$$\mathcal{W}_{t+1} \leftarrow \mathcal{W}_t - \gamma_t \left( g(\widetilde{\mathcal{W}_t^{(N:M)}}) + \lambda_W (\mathcal{W}_t - \widetilde{\mathcal{W}_t^{(N:M)}}) \right), \tag{9}$$

where $\lambda_W$ is a tunable hyperparameter, $\gamma_t$ is learning rate at time $t$, and $g$ is the gradient.

Traditional progressive sparse training works well for highly sparse model training (Han et al., 2015; Pool & Yu, 2021). However, its training data distribution does not change. According to the analysis of the two distribution shifts in Section 2.3, only one distribution shift is usually optimized when training using stochastic gradient descent or its improved algorithms. The optimization will fail if two distribution shifts are optimized simultaneously, such as switching the sparsity rate when training a DM. Directly training an extremely sparse model, such as 1:32 ($\frac{1}{32} = 0.03125$) sparsity, initialized by dense model weight and using extremely sparse weight gradients for reverse propagation may cause the sparse model to collapse. In this process, knowledge of the dense model is easily lost.

## 3.2 TRANSFER LEARN SPARSE DIFFUSION MODELS

To train a highly sparse DM from a dense DM, we fix the sparsity rate during DM training and transfer the knowledge to the sparse model via samples generated by the dense model. This method can accelerate the sparse training of DM and reuse the knowledge of the dense model as much as

possible. This method does not change the number of sampling steps used to generate samples. The following is a formal description of progressive knowledge transfer.

$x_0$ is the data used to train DM. During training, a sparse neural network $\mathcal{F}_{\mathcal{W}}(x_t, t)$ is trained to predict the noise in $x_t$ w.r.t. $x_0$ by minimizing the L2 loss between them. The loss is formulated as follows:

$$\mathcal{L}_{\text{diff}} := ||\epsilon_t - \mathcal{F}_{\mathcal{W}}(x_t, t)||_2^2, \tag{10}$$

The overall loss function of sparse training comprises the original dense task loss $\mathcal{L}_{\text{dense}}$, a diffusion loss $\mathcal{L}_{\text{diff}}$ that optimizes the diffusion model. It can be formulated as:

$$\mathcal{L}_{\text{sparse}}^i = \lambda_1 \mathcal{L}_{\text{dense}} + \lambda_2 \mathcal{L}_{\text{diff}}, \quad i \in [1, k], \tag{11}$$

where $\lambda_1$ and $\lambda_2$ are hyper-parameters to balance the losses, with range of $[0, 1]$, and $k$ is the sparse mask number. When training a sparse model, especially extremely sparse models, we will select a teacher model, such as a dense or sparse model, and adjust the values of $\lambda_1$ and $\lambda_2$.

To achieve 2:4 sparse GPU acceleration, the current sparse training method is to directly train from a dense model with a sparsity rate of 0 to a sparse model with a sparsity rate of $50\%$, which can easily cause information loss. Due to the lengthy training process aimed at obtaining $50\%$ of the sparse model, we adopt a progressive sparse training process, ensuring we can obtain $50\%$ of the sparse model with minimal information loss. We add progressive sparse masks to existing STE-based methods. The progressive sparse DM training task is expressed as follows:

$$\min_{S(\mathcal{W} \odot \mathcal{M})} \mathcal{L}(\mathcal{W}; \mathcal{D}; [\mathcal{M}_1, \mathcal{M}_2, ..., \mathcal{M}_k]), \tag{12}$$

The projection function $S(\cdot)$, which is non-differentiable during back-propagation, generates the $N{:}M$ sparse sub-network on the fly. To get gradients during back-propagation, STE computes the gradients of the sub-network $g(\widetilde{\mathcal{W}}) = \nabla_{\widetilde{\mathcal{W}}} \mathcal{L}(\widetilde{\mathcal{W}}; \mathcal{D})$ based on the sparse sub-network $\widetilde{\mathcal{W}}$, which can be directly back-projected to the dense network as the approximated gradients of the dense parameters. The approximated parameter update rule for the dense network can be formulated as:

$$\mathcal{W}_{t+1} \leftarrow \mathcal{W}_t - \gamma_t g(\widetilde{\mathcal{W}_t^{(N_i:M_i)}}), \quad i \in [1, k], \tag{13}$$

where $\gamma_t$ is learning rate at time $t$, $g$ is the gradient, $\widetilde{\mathcal{W}_t^{(N_i:M_i)}}$ is the sparse weight with mask $N_i{:}M_i$ at time $t$. This STE-based method could be easily improved by sparse mask regularization as follows:

$$\mathcal{W}_{t+1} \leftarrow \mathcal{W}_t - \gamma_t \left( g(\widetilde{\mathcal{W}_t^{(N_i:M_i)}}) + \lambda_W (\mathcal{W}_t - \widetilde{\mathcal{W}_t^{(N_i:M_i)}}) \right), \quad i \in [1, k], \tag{14}$$

where $\lambda_W$ is a tunable hyperparameter.

### 3.3 2:4 SPARSE MASK FOR SAMPLING ON GPU

The structured sparse function provides fully connected layers and convolutional layers with 2:4 sparse weights to achieve 2:4 sparse acceleration on the GPU. If their weights are pruned ahead of time, these layers can be accelerated using structured sparse functions on the GPU. To test real inference acceleration on an NVIDIA GPU, we used the 2:4 sparse operator in the acceleration library provided by NVIDIA. NVIDIA Ampere architecture GPUs have implemented CUDA operators to accelerate the multiplication of such matrices. 2:4 sparsity inference is expressed as:

$$y_t^{(2:4)} = \mathcal{F}(\widetilde{\mathcal{W}^{(2:4)}}; x_t), \tag{15}$$

where $x_t$ is input data at $t$, $y_t^{(2:4)}$ is the inference output of 2:4 sparse DM at time $t$. Transposable masks (Hubara et al., 2021) is one of the critical technologies for NVIDIA Ampere architecture GPUs to accelerate sparse matrices. Transposable masks suggest that a weight matrix and its transpose can be simply pruned by multiplying binary masks, so the backward pass shares the same sparse weight matrix with the forward pass.

## 4 RESULTS AND DISCUSSION

In this section, we will compare and discuss the different sparsity rate optimization results and some ablation study results.

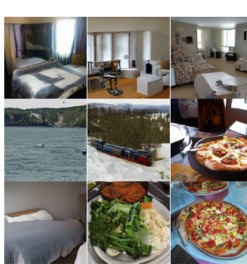 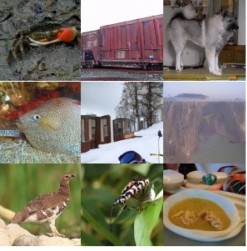 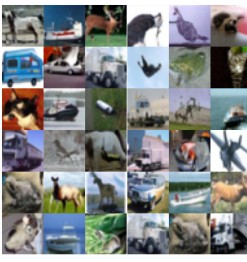 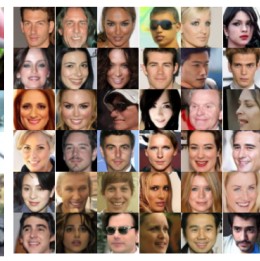

(a) MS-COCO 256×256    (b) ImageNet 256×256    (c) CIFAR10 32×32    (d) CelebA 64×64

Figure 2: Image generation results of 2:4 sparse U-ViT: selected samples on MS-COCO 256×256, ImageNet 256×256, on CIFAR10 32×32, and CelebA 64×64.

## 4.1 EXPERIMENTAL SETTINGS

**Evaluation Metrics.** In this paper, we concentrate primarily on two types of metrics: The efficiency metric is MACs; The quality metric is FID, which Equ. (16) calculates.

$$FID_{(x,g)} = \|\mu_x - \mu_g\|_2^2 + T_r(\sigma_x + \sigma_g - 2(\sigma_x\sigma_g)^{0.5}), \tag{16}$$

where $T_r$ represents the trace of the matrix. $x$ and $g$ represent real pictures and generated pictures, $\mu$ represents the mean, and $\sigma$ is the covariance matrix. We generated 50,000 images and calculated FID together with the original images.

**Baseline Methods.** We compare the performance of our method with the following baseline algorithm at 2:4 or other sparsity. Diff-Pruning (Fang et al., 2023b) is a method of structural pruning of diffusion models. ASP (Automatic SParsity) (Pool & Yu, 2021; Mishra et al., 2021) is a tool that enables 2:4 sparse training and inference for PyTorch models provided by Nvidia. Nvidia developed a simple training workflow that can easily generate a 2:4 structured sparse network that matches the accuracy of the dense network. STE-based Pruning (Bengio et al., 2013; Zhang et al., 2022) only uses sparse masks for training and inference. To compare the performance of other sparse masks, we give the results of inference with different sparsity rates. Finally, we provide the ablation study results of STE-based for the training process and traditional progressive sparse training. We give the ablation study results of untrained and STE-trained masks for the inference process.

## 4.2 RESULT COMPARISON

**2:4 Sparsity Results.** We experimented with four datasets (CIFAR10, CelebA, MS COCO 2014, and ImageNet) and three resolutions (32×32, 64×64, and 256×256) on Transformer-based and UNet-based DMs. We reduced the computational load by approximately 50%, and the FID only increased by 0.44 on average. The higher the resolution, especially for models of similar sizes, the better the FID effect, such as U-ViT on CIFAR10 32×32 and CelebA 64×64. This way, our method is more suitable for model acceleration in high-resolution and high-fidelity image generation.

In addition to changes in FID, it is also important to intuitively evaluate the data generated by the sparse acceleration model. From the generated images in Fig. 2, it can be seen that there is almost no difference between the images generated by our accelerated model and the images generated by the original model due to a slight change in FID.

Based on the theoretical acceleration results of 50%, Table 3 shows the actual acceleration results of testing the 2:4 sparse operator on the GPU, which is approximately 1.2x. The acceleration result is defined as the ratio between the running of a dense model and a sparse model in the same setting. The acceleration ratio should almost be the same across all datasets, and the weights have nothing to do with this. There are two significant hyper-parameters of the experiment: mlp_ratio, which sets the dimension of the MLP layer according to the attention layer dimension, and patch_size, which sets the sequence length for attention computation. Based on experimental experience, the ratio of sequence length to header dimension affects the speedup ratio. Some other hyper-parameters are head_dimension is 1024, num_head is 8, and depth is 1. In the above experiments, we only used the most primitive acceleration of 2:4 sparsity provided by PyTorch. Defining more advanced

Table 2: The comparison of 2: 4 (50% ) sparsity results. No Pruning is the method of U-ViT or DDPM. U-ViT with LDM is U-ViT using Latent Diffusion Models (LDM) (Rombach et al., 2022) for data process. Diff-Pruning is a structural pruning method. ASP is the method of sparse pruning from NVIDIA. STE-based Pruning (Bengio et al., 2013; Zhang et al., 2022) only uses sparse masks for training and inference. DDPM (Ho et al., 2020) is a diffusion model based on U-Net.

| Start DM | Datasets | Methods | FID | MACs (G) |
|---|---|---|---|---|
| U-ViT (Transformer-Based) | CIFAR10 32x32 | No Pruning | 3.97 | 11.34 |
| | | Diff-Pruning | 12.63 | 5.32 |
| | | ASP | 319.87 | 5.76 |
| | | STE-based Pruning | 4.23 | 5.67 |
| | | **Ours** | **3.81** | **5.67** |
| | CelebA 64x64 | No Pruning | 3.30 | 11.34 |
| | | Diff-Pruning | 11.35 | 5.32 |
| | | ASP | 438.31 | 5.76 |
| | | STE-based Pruning | 3.75 | 5.67 |
| | | **Ours** | **3.52** | **5.67** |
| U-ViT (Transformer-Based) with LDM | MS-COCO 256x256 | No Pruning | 5.95 | 11.34 |
| | | Diff-Pruning | 15.20 | 5.43 |
| | | ASP | 350.87 | 5.79 |
| | | STE-based Pruning | 8.14 | 5.68 |
| | | **Ours** | **7.09** | **5.68** |
| | ImageNet 256x256 | No Pruning | 3.81 | 76.66 |
| | | Diff-Pruning | 14.28 | 34.06 |
| | | ASP | 367.41 | 37.93 |
| | | STE-based Pruning | 5.83 | 36.84 |
| | | **Ours** | **5.25** | **36.84** |
| DDPM (UNet-Based) | CIFAR10 32x32 | No Pruning | 3.23 | 9.44 |
| | | Diff-Pruning | 12.31 | 4.75 |
| | | ASP | 328.47 | 4.91 |
| | | STE-based Pruning | 3.83 | 4.87 |
| | | **Ours** | **3.12** | **4.87** |
| | CelebA 64x64 | No Pruning | 2.94 | 9.44 |
| | | Diff-Pruning | 11.19 | 4.75 |
| | | ASP | 441.52 | 4.91 |
| | | STE-based Pruning | 3.66 | 4.87 |
| | | **Ours** | **3.04** | **4.87** |

Table 3: Speedup results for 2:4 (50% ) sparse model sampling on CIFAR10 32x32 data tested on 4 A40 GPUs. Patch_size and mlp_ratio are tunable hyperparameters.

| Acceleration | patch_size=1 | patch_size=2 | patch_size=4 |
|---|---|---|---|
| mlp_ratio=1 | 1.02 | 1.01 | 1.01 |
| mlp_ratio=2 | 1.04 | 0.97 | 1.01 |
| mlp_ratio=4 | **1.23** | 1.10 | 0.98 |
| mlp_ratio=8 | **1.22** | 1.17 | 1.08 |

acceleration operators, such as (Hu et al., 2024), provides further acceleration solutions. If we can integrate these methods into our model, our model can achieve faster training and inference.

**Other Sparsity Results.** To better understand the performance of our sparse training method, except for the 50% sparsity ratio of 2:4, we have also conducted experiments on other sparsity ratios. Since the ASP method is a sparsity tool provided by Nvidia for GPU hardware acceleration, no other sparsity ratios are provided. The following is mainly a comparison of different sparsity levels with other methods.

Fig. 3(a) shows the effect of different sparsity ratios on FID. We evaluated ten sparsity ratios with 32:32, 31:32, 15:16, 7:8, 3:4, 2:4, 1:4, 1:8, 1:16, and 1:32. As shown in this figure, it does not mean that the greater the sparsity, the better the FID. For example, the sparsity of 31:32 ($\frac{31}{32} = 0.96875$) and the sparsity of 15:16 ($\frac{15}{16} = 0.9375$), which is higher than the FID of 7:8 with a sparsity of

$(\frac{7}{8} = 0.875)$. In addition to GPU hardware acceleration, the sparsity ratio 2:4 also achieves good FID, proving the 2:4 sparse mask training and inference effectiveness.

Except for the 2:4 sparsity ratio, Fig. 3(c) shows that our method achieves significantly lower FID than the Diff-Pruning method at different sparse ratios on CIFAR10 and CelebA64.

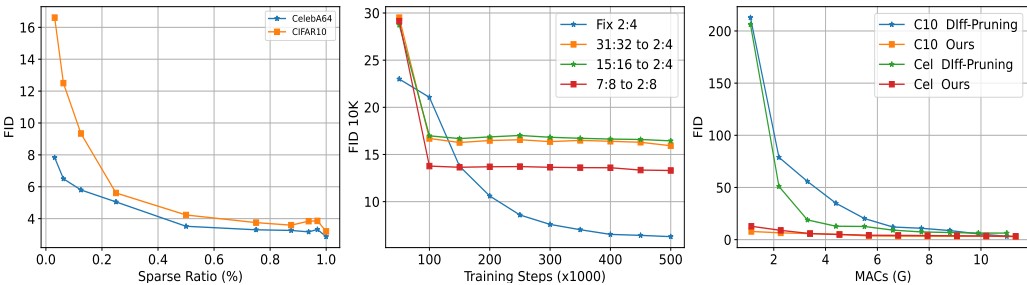

(a) The relationship between FID and mask sparsity ratio.

(b) Comparison between traditional progressive and fixed sparse training.

(c) Comparison of FID, MACs trade-off.

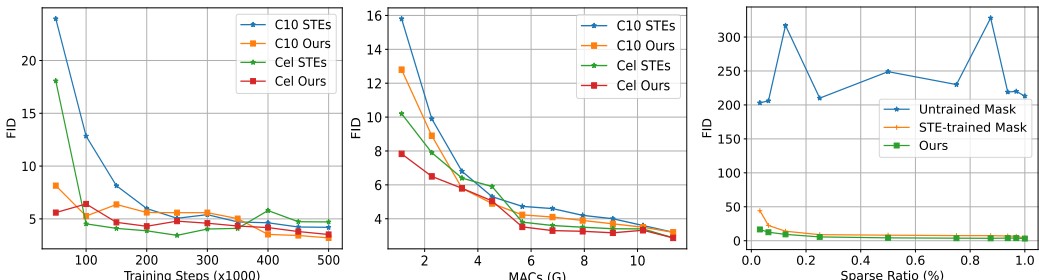

(d) Comparison of 2:4 sparse training process.

(e) Comparison of FID, MACs trade-off.

(f) Comparison between the untrained, STE-trained and our mask.

Figure 3: The comparison of sparsity results.

### 4.3 ABLATION STUDY

Our method involves performing sparse mask fine-tuning on the existing trained model. Masks need to be trained during the training process. We designed three ablation experiments to demonstrate better our method design's rationality and the necessity of each step. The first one is the STE-based method. The second one is traditional progressive sparse training of DM. The last one involves fixing the mask during the training process and not performing sparse training.

**STE Sparse Training.** As shown in Fig. 3(d), on datasets CIFAR10 and CelebA64, the learning curve of STE converges significantly slower than our method, mainly because our sparse model is trained with knowledge transfer. As shown in Fig. 3(e), STE's results are worse than ours, mainly because our method adds sparse mask information during back-propagation. At high compression ratios, such as 1:32, transfer learn sparse masks also plays an important role.

**Traditional Progressive Sparse Training.** As shown in Fig. 3(b), DM is trained on dataset CIFAR10. Training after switching sparse masks for the first time is almost ineffective. Every 100,000 training steps, the sparsity of masks is increased for progressive sparse training. The traditional progressive sparse training does not work well on the diffusion model, especially when switching the sparsity rate every time. It is equivalent to training stopped. However, the fixed sparse mask can be continuously trained until convergence. This comparison shows that simultaneously transforming the data and model distribution will fail DM training, prompting us to propose a new method for gradually sparsely training DMs.

**Sparse Mask Inference.** Unlike our trained masks, we generate untrained sparse masks for sparse pruning diffusion models. Dense model weights are imported into the sparse model, and the sparse model is not trained. As shown in Fig. 3(f), on dataset CIFAR10, the untrained mask's result is the worst at all MACs conditions because the mask did not participate in training. The STE-trained

mask is worse than ours because the mask did not transfer knowledge from dense model. Compared to the STE-trained masks, our method achieves 1 less FID at 9 sparsity rates.

## 5 RELATED WORK

### 5.1 N: M SPARSITY

A neural network with $N{:}M$ sparsity ($M$-weights contains $N$ non-zero values) satisfies that, in each group of $M$ consecutive weights of the network, there are at most $N$ weights have non-zero values (Zhou et al., 2021; Chmiel et al., 2022). APEX's Automatic SParsity (ASP) (Pool & Yu, 2021; Mishra et al., 2021)) is a 2:4 sparse tool provided by NVIDIA. This tool obtains a 2:4 sparse network and can achieve nearly 2x runtime speedup on NVIDIA Ampere and Hopper architecture GPUs (Nvidia, 2020). A provable and efficient method for finding N: M transposed masks for accelerating sparse neural training (Hubara et al., 2021). Compared with dense networks, sparse network training has gradient changes, and methods such as STE (Bengio et al., 2013) should be used to improve training performance. In the forward stage, sparse weight is obtained by pruning dense weight. In the backward stage, the gradient w.r.t. sparse weight will be directly applied to dense weight. SR-STE (Zhang et al., 2022) enhances the backward pass information by integrating the sparse weight of the forward pass into the backward pass. DominoSearch (Sun et al., 2021) found mixed N: M sparsity schemes from pre-trained dense, deep neural networks to achieve higher accuracy than the uniform-sparsity scheme with equivalent complexity constraints. The N: M sparse method is also used in SparseGPT (Frantar & Alistarh, 2023) to accelerate the LLM generation model GPT. Although many N: M sparse methods exist, none have been proven effective for sparse DM.

### 5.2 DIFFUSION MODEL PRUNING

Pruning is one of the most used methods to reduce the calculation time of DNN, including DM. The pruning method was divided into structured pruning and unstructured pruning. Structural pruning aims to physically remove a group of parameters, thereby reducing the size of neural networks. In contrast, unstructured pruning involves zeroing out specific weights without altering the network structure (Fang et al., 2023a). Sparsity can reduce the memory footprint of DM to fit mobile devices and shorten training time for ever-growing networks (Hoefler et al., 2021). Pay attention to features and selection of useful features for the target dataset (Wang et al., 2020), which can also reduce computational complexity. The parameter-grouping patterns vary widely across different models, making architecture-specific pruners, which rely on manually designed grouping schemes, non-generalizable to new architectures. Depgraph (Fang et al., 2023a) can tackle general structural pruning of arbitrary architecture. Structured sparsity was also used for large language models (Ma et al., 2023; Frantar & Alistarh, 2023; Guo et al., 2023). Because Transformer has been proven to outperform the other networks in multiple applications, including DM. ToMe (Bolya & Hoffman, 2023) merge redundant tokens of Transformer. LD-Pruner (Castells et al., 2024) and P-ESD (Yang et al., 2024) prune stable diffusion for specific tasks. There are also proposals for early exiting for accelerated inference in DM (Moon et al., 2023). Although there are many pruning methods for DM, either the sparse model FID is poor or structured 2:4 sparsity is not implemented for general base DM.

## 6 CONCLUSION

In this paper, we studied how to improve the efficiency of DM by sparse matrix for 2:4 sparse acceleration GPU. The existing STE-based methods make it difficult to optimize sparse DM. To address this issue, we improve the STE method and propose to gradually transfer knowledge from dense models to sparse models. Our method is tested on the latest Transformer-based DM, U-ViT, and UNet-based DM, DDPM. We trained the 2:4 and other sparse models to perform better than other methods. Our approach also provides an effective solution for DM deployment on NVIDIA Ampere architecture GPUs, achieving about 1.2x acceleration.

# 7    ETHICS STATEMENT

While our motivation is to make diffusion models generate images faster, we also recognize the ethical implications inherent in model pruning, as sparse optimization may introduce unintended biases. There are differences between traditional evaluation metrics (such as FID scores) and human eye evaluation, and optimization for FID may not capture subtle deviations in human vision. In addition to performance improvements to diffusion models, we are also actively focusing on the ethical aspects of AI deployment. We are committed to preventing the spread of content that may have a negative social impact. Prevent unintended consequences by acknowledging the limitations of FID indicators.

# 8    REPRODUCIBILITY

**Implementation Details.** All of the experiments are implemented by PyTorch and conducted on servers with 24 GPUs, eight NVIDIA 4090, eight A40, and eight 3090 GPUs. We evaluate the FID score every 50K training iterations on 10K generated samples (instead of 50K samples for efficiency).

The anonymous address of the source code for the paper is https://github.com/aaa-bbb-111/SparseDM. The paper code implementation consists of two parts. The first part, sparse training, trains the sparse DM and verifies the acceleration theory of the sparse diffusion model. The second part, GPU inference, verifies GPU inference acceleration.

# A    APPENDIX

## A.1    LIMITATIONS AND FUTURE WORK

The MACs of the sparse diffusion model are significantly reduced, but the acceleration effect on real processors is limited, especially the 2:4 sparse acceleration that can currently only reach about 1.2 times. Future work will design sparse acceleration schemes for different processors to achieve better acceleration.

## A.2    PARAMETER COUNTS FOR EACH LAYER OF THE DM BEFORE AND AFTER SPARSE PRUNING

We add a 2:4 sparse mask to each convolutional and fully connected layer, so that all models have 50% of the parameters of each convolutional and fully connected layer. For example, the 5th fully connected layer of U-ViT has 600,000 parameters, which becomes 300,000 parameters after sparse masking. We have added this explanation in the revised paper.

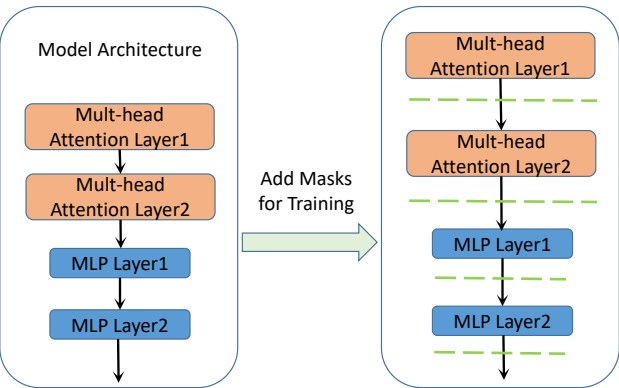

Figure 4: Add sparse mask to each layer

### A.3 THE RELATIONSHIP BETWEEN DIFFUSION TRAINING AND SPARSE TRAINING

As shown in Fig. 3(b) of this paper, from the empirical experimental results, it is observed that fixed sparsity applies a consistent distribution shift for all noise levels in diffusion training, while progressive sparsity training gradually shifts the predefined noise level, which may hinder the diffusion training process.

In theory, the relationship between diffusion training and sparse training is mainly explained from the perspective of the difficulty of convergence of the SGD optimizer. Existing optimizers are designed for diffusion training and sparse training, respectively, and the design of each optimizer is challenging. SGD takes the current optimal gradient direction each time it descends, so when using stochastic gradient descent training, usually only one distribution shift is optimized. However, if there are two distribution shifts, the SGD gradient descent direction may not be the current optimal one. Therefore, if two distribution shifts are optimized at the same time, such as switching the sparsity rate when training DM, the optimization will fail.

### A.4 DENSE AND SPARSE MATRIX ON GPU

As shown in Fig. 5: Dense and sparse matrices on the GPU, in order to implement the sparse structure, Nidia CUDA defines an additional 2-bit indices matrix for calculation. Therefore, when the sparse matrix is not large enough, this additional indices matrix overhead will make the overall result slower. Therefore, on the GPU, the larger the network model, the better the acceleration results may be. Fig. 5 is from Google Image. https://images.app.goo.gl/7CDgZVcuUYG8rzyc6

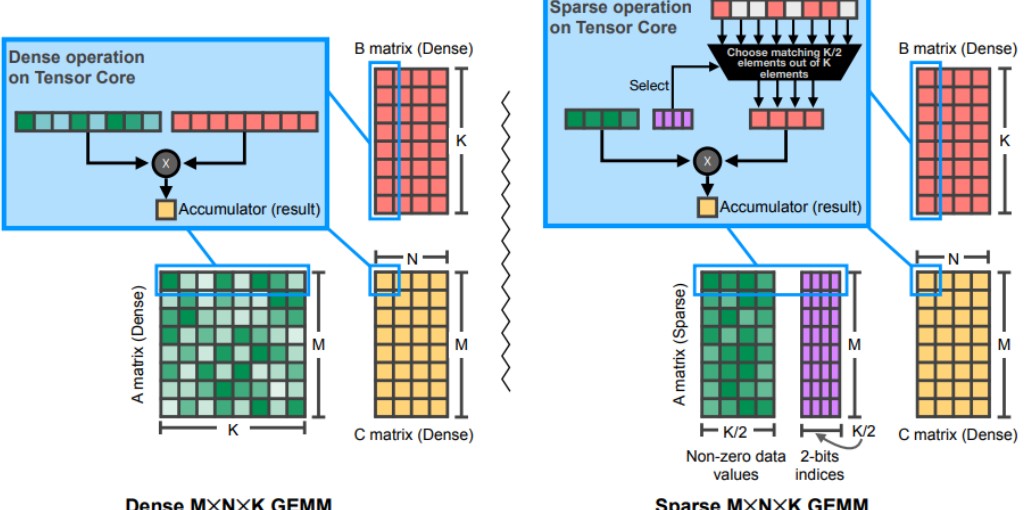

Figure 5: Dense and sparse matrix on GPU.

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
