# OpenReview forum: "SparseDM: Toward Sparse Efficient Diffusion Models"
_ICLR.cc/2025/Conference — ICLR 2025 Conference Withdrawn Submission_

### Official Review · Reviewer_X5E7 · 2024-10-20

**Soundness:** 2
**Presentation:** 2
**Contribution:** 3
**Rating:** 5
**Confidence:** 4

**Summary:**

This paper introduces SparseDM, which converts existing diffusion models into sparse models that fit in a 2:4 sparse operator on the GPU. Specifically, the authors propose a Straight-Through Estimator (STE)-based fine-tuning framework that learns sparse masks. These sparse masks accelerate GPU inference speed up to 1.2. Comprehensive experiments validate the effectiveness of the proposed method.

**Strengths:**

* The paper introduces a simple fine-tuning method that converts existing diffusion models into sparse models, enabling them to be used in scenarios with limited computing power, such as on mobile devices.

* The observations about fixed sparse training are interesting.

* Experiments on various generation scenarios verify the effectiveness of SparseDM compared to baselines.

**Weaknesses:**

**Weakness 1: More clarifications on Section 2.3.**

In Section 2.3, the authors claim that diffusion models only consider the distribution shift of the noisy data while sparse pruning methods only consider the model's weight change. Then, referring to RFR, the authors convert the model's weight changes resulting from sparse pruning methods into data changes for the diffusion model's training process. However, typical diffusion models have indicators for perturbed data (such as the noise schedule and timestep embedding), and it is unclear how these relate to perturbations caused by sparse training.

**Weakness 2: Lack of analysis of fixed sparse training**

I am not sure why fixed sparse training would be more effective than traditional progressive sparse training. Based on the experimental results, it seems that fixed sparsity applies a consistent distribution shift across all noise levels in diffusion training, whereas progressive sparse training gradually shifts the predefined noise levels, which may hinder the diffusion training process. However, this claim has not been theoretically verified, so the authors should provide theoretical proof to demonstrate the relationship between diffusion training and sparse training.

**Questions:**

* In Table 3, some variants (e.g., patch size = 2 and mlp_ratio = 2) are slower than the dense model, why do you think this is?
* I think it would strengthen the effectiveness of SparseDM if the author show that it can also be applied to models like Stable Diffusion.

---

> ### Author Response · Authors · 2024-11-27
> **Response to Reviewer X5E7**
>
> Dear Reviewer X5E7,
>
> Thank you for your comments and constructive review. Below we address the concerns and questions mentioned.
>
> **W1:** More clarifications on Section 2.3.
> In Section 2.3, the authors claim that diffusion models only consider the distribution shift of the noisy data while sparse pruning methods only consider the model's weight change. Then, referring to RFR, the authors convert the model's weight changes resulting from sparse pruning methods into data changes for the diffusion model's training process. However, typical diffusion models have indicators for perturbed data (such as the noise schedule and timestep embedding), and it is unclear how these relate to perturbations caused by sparse training.
>
> **AW1:** Diffusion models have indicators of data perturbations (e.g., noise schedules and time-step embeddings) that may coexist with model perturbations caused by sparse training.
> As shown in Figure 3(b) of this paper, experimental observations show that switching the mask sparsity rate causes training to stagnate and fail to converge. Switching the mask sparsity rate during sparse diffusion model training also makes it difficult to reuse dense model knowledge.
>
>
> **W2:** Lack of analysis of fixed sparse training
> I am not sure why fixed sparse training would be more effective than traditional progressive sparse training. Based on the experimental results, it seems that fixed sparsity applies a consistent distribution shift across all noise levels in diffusion training, whereas progressive sparse training gradually shifts the predefined noise levels, which may hinder the diffusion training process. However, this claim has not been theoretically verified, so the authors should provide theoretical proof to demonstrate the relationship between diffusion training and sparse training.
>
> **AW2:** As shown in Figure 3(b) of this paper, from the empirical experimental results, it is observed that fixed sparsity applies a consistent distribution shift for all noise levels in diffusion training, while progressive sparsity training gradually shifts the predefined noise level, which may hinder the diffusion training process.
>
> In theory, the relationship between diffusion training and sparse training is mainly explained from the perspective of the difficulty of convergence of the SGD optimizer. Existing optimizers are designed for diffusion training and sparse training, respectively, and the design of each optimizer is challenging. SGD takes the current optimal gradient direction each time it descends, so when using stochastic gradient descent training, usually only one distribution shift is optimized. However, if there are two distribution shifts, the SGD gradient descent direction may not be the current optimal one. Therefore, if two distribution shifts are optimized at the same time, such as switching the sparsity rate when training DM, the optimization will fail.
>
> **Q1:** In Table 3, some variants (e.g., patch size = 2 and mlp\_ratio = 2) are slower than the dense model, why do you think this is?
>
> **A1:** This Google image shows dense and sparse matrices on a GPU.
>
> https://images.app.goo.gl/7CDgZVcuUYG8rzyc6
>
> In order to implement the sparse structure, Nidia CUDA defines an additional 2-bit indices matrix for calculation. Therefore, when the sparse matrix is not large enough, this additional indices matrix overhead will make the overall result slower.
> Therefore, on the GPU, the larger the network model, the better the acceleration results may be.
>
> **Q2:** I think it would strengthen the effectiveness of SparseDM if the author show that it can also be applied to models like Stable Diffusion.
>
> **A2:** Thank you very much for your suggestions for improvement. At present, the sparse acceleration on SD is still being optimized, and the results of SD sparse acceleration will be announced in subsequent work.
>
> We are performing sparse acceleration on the general basic diffusion model, so it has been verified to be effective on the diffusion models based on Transformer and U-Net.
> The basic diffusion model of SD 1.5 is also based on U-Net, and the basic diffusion model of SD 3.0 is also based on Transformer.
> Therefore, our method can be generalized to SD.
> To prevent the influence of multi-modal models, such as CLIP, in SD, we currently only perform experiments on the diffusion models, U-ViT and DDPM.

---

### Official Review · Reviewer_Q9z9 · 2024-10-30

**Soundness:** 1
**Presentation:** 3
**Contribution:** 1
**Rating:** 3
**Confidence:** 4

**Summary:**

This work aims to reduce the computation of Diffusion Models during inference. The authors suggest a method of straight-through estimation, which applies sparse masks to layers of a pretrained diffusion model and then employs transfer learning for training. Then, they use the same sparse mask during inference to improve compute efficiency.

**Strengths:**

- The 2:4 sparse model calculation offers practical values for practitioners using NVIDIA Ampere architecture GPUs.

**Weaknesses:**

- While it may have some practical value for practitioners using NVIDIA Ampere architecture, the same technique may not benefit other practitioners or general researchers without access to Ampere architecture.

- Besides, the straightforward idea of using masked training is neither interesting nor technically new.

- More disappointingly, the speed acceleration due to this customized training for a particular architecture increases by x1.2 only. Studies related to reducing time steps for Diffusion inference or diffusion quantization/pruning methods may be more effective in achieving the same purpose.

**Questions:**

Please address the weakness stated above.

---

> ### Author Response · Authors · 2024-11-26
> **Response to Reviewer Q9z9**
>
> Dear Reviewer Q9z9,
>
> Thank you for your comments and constructive review. Below we address the concerns and questions mentioned.
>
>
> **W1:** While it may have some practical value for practitioners using NVIDIA Ampere architecture, the same technique may not benefit other practitioners or general researchers without access to Ampere architecture.
>
> **A1:** Thank you for your confirmation that our work can be used on the NVIDIA Ampere architecture.
> Both NVIDIA Ampere and Hopper architectures support sparse matrix acceleration, such as A100 and H100.
>
> https://developer.nvidia.com/blog/accelerating-inference-with-sparsity-using-ampere-and-tensorrt/
>
> https://www.advancedclustering.com/wp-content/uploads/2022/03/gtc22-whitepaper-hopper.pdf
>
> https://arxiv.org/html/2402.13499v1
>
> To achieve sparse matrix acceleration on other hardware, such as FPGA, specialized sparse operators are needed.
>
> **W2:** Besides, the straightforward idea of using masked training is neither interesting nor technically new.
>
> **A2:** To the best of our knowledge, we are the first to use a sparse training pruning method on diffusion models.
>
> Our transfer learning sparse diffusion model strategy is significantly different from the existing masked training strategy.
> There are two main differences.
> 1. Our method must perform progressive sparsity across models.
> 2. Our method must transfer knowledge across models.
> These two improvements are aimed at solving the problem that the existing masked training strategy fails in the diffusion model.
>
> **W3:** More disappointingly, the speed acceleration due to this customized training for a particular architecture increases by x1.2 only. Studies related to reducing time steps for Diffusion inference or diffusion quantization/pruning methods may be more effective in achieving the same purpose.
>
> **A3:** Model quantization and time step reduction can achieve more than 1.2 times speedup, but these methods are based on dense matrix model acceleration, and our matrix sparse pruning is orthogonal to these two methods.
>
> The current SD pruning methods, such as "Pruning for Robust Concept Erasing in Diffusion Models" and "LD-Pruner: Efficient Pruning of Latent Diffusion Models using Task-Agnostic Insights" are a speedup after fine-tuning for specific tasks.
> Our method is a general base diffusion model acceleration. It is not targeted at a specific task.
> We are performing sparse acceleration on the general basic diffusion model, so it has been verified to be effective on the diffusion models based on Transformer and U-Net.
> The basic diffusion model of SD 1.5 is also based on U-Net, and the basic diffusion model of SD 3.0 is also based on Transformer.
> Therefore, our method can be generalized to SD.
>
> So, hopefully, you will consider our contribution to the acceleration of the sparse diffusion model.

---

> > ### Comment · Reviewer_Q9z9 · 2024-11-28
> >
> > Regarding A1: Given the scenarios where efficient diffusion models are most needed, the focus should be on solutions that perform well on edge devices with limited computational capacity, rather than relying on high-capacity processors like A100 or H100. How does this method address efficiency in resource-constrained environments?
> >
> > Regarding A2: While pruning methods were discussed, recent works like Structural Pruning for Diffusion Models (Arxiv, 2023) and LD-Pruner (CVPR 2024 Workshop) demonstrate superior computational reduction compared to the proposed approach. I disagree that the proposed work is the first method that uses a pruning method on diffusion models.
> >
> > Regarding A3: The authors emphasize the generalizability of their approach to downstream applications. However, the method’s utility appears to be tied to specific hardware (A100/H100). If generalization is the goal, addressing hardware dependence should be a priority. Alternatively, if performance is the focus, the rebuttal needs to convincingly demonstrate superiority over existing methods with similar goals.
> >
> >
> > The rebuttal leaves ambiguity about the true strengths of the proposed method compared to prior work. What unique contribution does this method bring beyond hardware-specific optimizations? A clearer articulation of its advantages in comparison to recent research is necessary for a stronger case.

---

> > > ### Author Response · Authors · 2024-11-28
> > > **Response to Reviewer Q9z9**
> > >
> > > Dear Reviewer Q9z9,
> > >
> > > Thank you for your comments.
> > >
> > > **Regarding A1:** Given the scenarios where efficient diffusion models are most needed, the focus should be on solutions that perform well on edge devices with limited computational capacity, rather than relying on high-capacity processors like A100 or H100. How does this method address efficiency in resource-constrained environments?
> > >
> > > **AA1:** Based on current technology, A100 and H100 are rarely used in edge devices. However, A100 and H100 may be used in cars, which is also a scenario with limited computing resources. Our accelerated inference method can reduce the inference cost of large-scale diffusion models (such as Sora) in cloud computing.
> > >
> > > **Regarding A2:** While pruning methods were discussed, recent works like Structural Pruning for Diffusion Models (Arxiv, 2023) and LD-Pruner (CVPR 2024 Workshop) demonstrate superior computational reduction compared to the proposed approach. I disagree that the proposed work is the first method that uses a pruning method on diffusion models.
> > >
> > > **AA2:** Structural Pruning for Diffusion Models (Arxiv, 2023) and LD-Pruner (CVPR 2024 Workshop) perform better on certain specific tasks. Our method is not a specific task acceleration on SD but a sparse acceleration for the general basic diffusion model. Our sparse acceleration method does not require re-fine-tuning for a specific task.
> > >
> > > **Regarding A3:** The authors emphasize the generalizability of their approach to downstream applications. However, the method’s utility appears to be tied to specific hardware (A100/H100). If generalization is the goal, addressing hardware dependence should be a priority. Alternatively, if performance is the focus, the rebuttal needs to convincingly demonstrate superiority over existing methods with similar goals.
> > >
> > > **AA3:** We are very sorry that we did not conduct sufficient research before. The Nvidia GPUs that support the 2:4 sparse acceleration operator include RTX3090, A40, L40, A100, H100, etc., which are common computing cards on the market.
> > > Therefore, our acceleration algorithm can be applied to various scenarios such as personal computers and cloud servers.
> > >
> > > So our method is not a hardware-specific optimization, but a general acceleration method.

---

> ### Comment · Reviewer_Q9z9 · 2024-12-03
>
> - The authors acknowledge that their proposed method cannot be applied to edge devices.
>
> - They claim that the generality of their approach lies in its applicability to various high-end GPUs without relying on specific tasks. However, I disagree with this argument. For practical deployment, methods should naturally be optimized on task-specific models to better align with real-world applications. While the unconditional diffusion models serve as a valuable baseline for various conditional generation tasks, all successful real-world diffusion models are designed with specific tasks in mind. Any compression method (intended for practical deployment) should demonstrate its applicability to clear use cases and serviceable scenarios.
>
> - Regarding the claim that the method can be added to other task-specific techniques, I strongly disagree. Introducing a conditional branch to a diffusion model significantly alters the distribution of activations and weights. Many model compression techniques are specifically designed to account for such changes. How can the authors confidently assert that their method will yield the same performance benefits (i.e., 1.2x performance gain without sacrificing the quality) when combined with these techniques? This claim must be supported by experimental validation through real-world applications.
>
> - I am also worried that the authors only recently confirmed the specific devices on which their method can be applied. Besides, even if hardware acceleration supports it, it only obtains a 20% gain in performance, isn't this too marginal?
>
>
> In conclusion, I recommend that the authors carefully reevaluate the practical use cases and value of their study. After substantial improvement, they could consider resubmitting to another conference. My recommendation remains the same: reject.

---

### Official Review · Reviewer_F6DV · 2024-10-31

**Soundness:** 3
**Presentation:** 3
**Contribution:** 2
**Rating:** 5
**Confidence:** 4

**Summary:**

This paper proposes to improve the efficiency of DM by sparse matrix for 2:4 sparse acceleration GPU. The authors improve the STE method and propose to gradually transfer knowledge from dense models to sparse models.

**Strengths:**

1.	This paper is well-written.
2.	The motivation is clear enough.
3.	The organization of this paper is great.

**Weaknesses:**

1.	There is a typo in Eq5. Please also check all equations. Moreover, not all symbols have been explained.
2.	The experiments are relatively limited. Specifically, only two U-ViT and DDPM are tested on the proposed pruning, which are proposed in 2022 and 2020 respectively. More recently proposed DiT or other methods should also be included.
3.	The limitation and discussion are missing in this paper.

**Questions:**

1.	The authors mentioned that “it does not mean that the greater the sparsity, the better the FID”. Please discuss the reason and why you choose 2:4 sparse.
2.	Please discuss the reason ASP performs so worse in all experiments.
3.	Please also clarify why your method and STE-based pruning fulfill the same MACs.
4.	Please explain the reason that the FID of the proposed method in Fig. 3a obtain a lower FID in the first several steps.
5.	Why the initial FID of 2:4 sparse in Fig.3b and Fig.3d is different?

---

> ### Author Response · Authors · 2024-11-26
> **Response to Reviewer F6DV**
>
> Dear Reviewer F6DV,
>
> Thank you for your comments and constructive review. Below we address the concerns and questions mentioned.
>
> **W1:** There is a typo in Eq5. Please also check all equations. Moreover, not all symbols have been explained.
>
> **AW1:** Thank you for helping us to carefully check the mathematical expressions and their symbolic interpretation. We have addressed minor issues on revised paper.
>
> **W2:** The experiments are relatively limited. Specifically, only two U-ViT and DDPM are tested on the proposed pruning, which are proposed in 2022 and 2020 respectively. More recently proposed DiT or other methods should also be included.
>
> **AW2:** So far, DiT (arXiv 2022, ICCV 2023) is used more than U-ViT (CVPR 2023).
> However, U-ViT and DiT are similar technologies, both of which are diffusion models implemented based on Transformer, and both have been officially published at top conferences.
> Both U-ViT and DiT are adopted by diffusers as open source basic diffusion models.
>
> https://huggingface.co/docs/diffusers/en/api/models/dit_transformer2d
>
> https://huggingface.co/docs/diffusers/en/api/models/uvit2d
>
> U-ViT is also the base model of the universal multi-modal diffusion model UniDiffuser (ICML 2023). UniDiffuser is a unified diffusion framework to fit all distributions relevant to a set of multi-modal data in one model.
>
> Therefore, we present experimental results of U-ViT that are sufficient to demonstrate that our sparse pruning technique can be applied to advanced diffusion models based on Transformer.
>
> We are performing sparse acceleration on the general basic diffusion model, so it has been verified to be effective on the diffusion models based on Transformer and U-Net.
> The basic diffusion model of SD 1.5 is also based on U-Net, and the basic diffusion model of SD 3.0 is also based on Transformer.
> Therefore, our method can be generalized to SD.
>
> **W3:** The limitation and discussion are missing in this paper.
>
> **AW3:** In Appendix A of the submitted paper, we point out some limitations and future work.
>
> Limitations and Future Work. The MACs of the sparse diffusion model are significantly reduced, but the acceleration effect on real processors is limited, especially the 2:4 sparse acceleration that can currently only reach about 1.2 times. Future work will design sparse acceleration schemes for different processors to achieve better acceleration.
>
> **Q1:** The authors mentioned that “it does not mean that the greater the sparsity, the better the FID”. Please discuss the reason and why you choose 2:4 sparse.
>
> **A1:** In our paper, Fig. 3(a) shows the effect of different sparsity ratios on FID.
> Dense model (sparsity 32:32) or high sparsity model (sparsity 31:32, 15:16) have a certain degree of redundancy and even overfitting. Introducing an appropriate amount of sparsity (sparsity 7:8) as a regularization term during model training can prevent the model from overfitting the data. A model with an appropriate amount of sparsity (sparsity 7:8) can achieve better FID.
>
> Since the best hardware currently supporting sparse matrix acceleration is Nvidia's Ampere and Hopper architectures (A100, H100), which only support 2:4 sparse matrix acceleration, we choose 2:4 sparse acceleration.
>
> **Q2:** Please discuss the reason ASP performs so worse in all experiments.
>
> **A2:** The Nvidia ASP prunes along channel dimensions, and our STE-based method prunes alone kernel dimensions.
> Our technical route is similar to the scheme "N:M fine-grained structured sparse neural network".
>
> https://github.com/aojunzz/NM-sparsity
>
> Our pruning technique is more sophisticated and less damaging to the model.
>
> **Q3:** Please also clarify why your method and STE-based pruning fulfill the same MACs.
>
> **A3:** Our methods include 2 parts: the first one is the improved STE method; the second is to transfer learning knowledge from the dense model.
> In our paper, STE-based pruning is performed without using the second part and has the same MACs as our final method, but worse FID.
>
> **Q4:** Please explain the reason that the FID of the proposed method in Fig. 3a obtain a lower FID in the first several steps.
>
> **A4:** Fig. 3(a) is not the training process of a model, but the trade-off between FID50000 and sparsity ratio for a fixed sparse training model.
> There are 9 models with different sparsity rates and 1 dense model and their corresponding FIDs.
>
> **Q5:** Why the initial FID of 2:4 sparse in Fig.3b and Fig.3d is different?
>
> **A5:** During the experiment, in order to quickly evaluate the intermediate model, we will use 10,000 images to calculate FID10000. Figure 3b is FID10000.
> Since the FID10000 of the intermediate models vary greatly, to check the convergence process of our method, we compute the FID50000 of some intermediate models with 50,000 images. Figure 3d is FID50000.
>
> We hope you will consider our contribution to sparse and efficient diffusion models.

---

### Official Review · Reviewer_QPmA · 2024-11-04

**Soundness:** 2
**Presentation:** 3
**Contribution:** 2
**Rating:** 3
**Confidence:** 3

**Summary:**

This paper proposes a pruning strategy for Diffusion models, using mask pruning to achieve progressive multi-step pruning. Ultimately, it realizes 1:2 pruning according to the Ampere architecture. During training, knowledge distillation is used to transfer knowledge from the full model to the pruned model.

**Strengths:**

The writing is very clear, and the main idea is highlighted effectively.

**Weaknesses:**

1. The pruning strategy is based on existing structures, with a relatively simple motivation. There are already other methods that achieve similar results, such as using linear attention or directly training a smaller model with distillation.
2. Compared to directly using STE-based pruning, it does not further reduce the computational load.
3. In Section 3.2, "Transfer learn sparse diffusion models" strategy is mentioned, but it does not explain the significant differences between this strategy and the progressive sparse training strategy discussed in Section 2.2. If the focus is solely on testing with perturbed datasets, it may not constitute a significant contribution.
4. A generalized pruning strategy suitable for Transformer networks has not been proposed; simply relying on data perturbations is insufficient to demonstrate applicability to other datasets. Further testing on additional datasets, such as CelebA-HQ, LSUN Church, would be beneficial.
5. Many of the latest comparative algorithms from 2024 are not mentioned, such as "Pruning for Robust Concept Erasing in Diffusion Models" and "LD-Pruner: Efficient Pruning of Latent Diffusion Models using Task-Agnostic Insights."
6. There is no comparison of the parameter counts for each layer of the SD model before and after sparse pruning. It is recommended to include a chart in the appendix to illustrate this.
7. While Section 2.3 mentions applying perturbations to the dataset, it does not provide specific details on how the perturbations were implemented.
8. The experiments only validate the FID score as a single metric; it is advisable to explore additional metrics, such as SSIM.

**Questions:**

Could the authors provide the parameter counts for each layer of the SD model before and after sparse pruning?

---

> ### Author Response · Authors · 2024-11-27
> **Response to Reviewer QPmA**
>
> Dear Reviewer QPmA,
>
> Thank you for your comments and constructive review. Below we address the concerns and questions mentioned.
>
> **W1:** ...
>
> **AW1:** To the best of our knowledge, we are the first to use sparse training pruning methods on diffusion models.
> Linear attention or directly training a smaller model with distillation can achieve speedup, but these methods are based on dense matrix model acceleration, and our matrix sparse pruning is orthogonal to these two methods.
> We hope you will consider our contribution to the acceleration of the diffusion model.
>
> **W2:** ...
>
> **AW2:** Our sparse pruning can achieve high sparsity, such as 1:32, so sparsity is not the bottleneck of our method.
> Compared with the vanilla STE pruning method, we focus on the FID performance of the model under the same MACs.
>
> Our methods include 2 parts: the first one is the improved STE method; the second is to transfer learning knowledge from the dense model.
> In our paper, STE-based pruning is performed without using the second part and has the same MACs as our final method, but worse FID.
>
> **W3:** ...
>
> **AW3:** Progressive sparse training strategy is to switch sparse masks during training, which is to improve the sparsity rate of masks in the same model. The knowledge reuse of dense models is completed in the same model.
> Our strategy for transferring training sparse models is to use different sparsity rates in different diffusion models for training, and our knowledge transfer is through transferring across models with different sparsity rates.
>
> So there are two obvious differences between our method and progressive sparsity:
> 1. Our method must perform progressive sparsity across models;
> 2. Our method must transfer knowledge across models.
>
> The progressive sparse training strategy is an ineffective optimization for changing the sparsity rate in the same diffusion model.
> As shown in Figure 3(b) of this paper, DM is trained on the CIFAR10 dataset. The training after the first switching of the sparse mask is almost ineffective.
> It was widely used before because it is effective for changing the sparsity rate in the same CNN model.
>
> The progressively sparse training strategy is also ineffective for knowledge reuse within the same diffusion model, which must be achieved by minimizing the loss of noisy predictions for both dense and sparse models, as shown in Equation (11) in our paper.
> For knowledge reuse in CNN models, just load the dense model weights and then finetune.
> So in essence, the existing progressive sparse training strategy can only be applied to CNN models and is invalid for diffusion models. We redesigned a set of transfer learning sparse model strategies for diffusion models.
>
> **W4:** ...
>
> **AW4:** CelebA-HQ and LSUN Church are datasets with 256x256 resolution. In Table 2 of our paper, we show that our method outperforms other methods on MS-COCO 256x256 and ImageNet 256x256.
>
> **W5:** ...
>
> **AW5:** Thanks for finding the paper on pruning SD.
>
> The paper "Pruning Robust Concept Erasure in Diffusion Models" takes a different view than ours.
> In our paper, we do not perform sparse training on multi-modal SD, but only model the process of adding and removing Gaussian noise to images.
> We do not edit images. Our research is to train a general sparse diffusion model, not for a specific task.
>
> The paper "LD-Pruner: Efficient Pruning of Latent Diffusion Models using Task-Agnostic Insights" uses a small dataset CelebA-HQ 256×256 to fine-tune the SD model knowledge, and then tests the FID and acceleration results on this small dataset.
> Theoretically, our method can train SD from scratch and then obtain a 1.2x actual acceleration in all common SD scenarios while maintaining the performance of dense SD models.
> Our method is a general base model acceleration. LD-Pruner is an acceleration after fine-tuning for a specific task.
>
> We cite the above two articles in the revised paper.
>
> **W6:** ...
>
> **AW6:** We add a 2:4 sparse mask to each convolutional and fully connected layer, so that all models have 50%
> of the parameters of each convolutional and fully connected layer.
> For example, the 5th fully connected layer of U-ViT has 600,000 parameters, which becomes 300,000 parameters after sparse masking. We have added this explanation in the revised paper.
>
> **W7:** ...
>
> **AW7:** In this paper, we argue that diffusion model training adds Gaussian noise to the data, which perturbs the original image data distribution. We explain this in the revised paper.
>
> **W8:** ...
>
> **AW8:** The FID metric has limitations in terms of generated image quality, but current papers do use only FID to evaluate image generation quality, such as “Pruning for Robust Concept Erasure in Diffusion Models” and “LD-Pruner: Efficient Pruning of Latent Diffusion Models Using Task-Agnostic Insights”.
>
> **Q1:** ...
>
> **A1:** Same as **AW6**

---

> > ### Comment · Reviewer_QPmA · 2024-12-03
> >
> > Regarding A1
> > The purpose of pruning is to accelerate model inference speed, but the method proposed in this paper heavily relies on the specific architecture design and does not present a general pruning technique. It is only applicable to the NVIDIA Ampere architecture.
> >
> > Regarding A7
> > The comparison between the progressive pruning strategy and the fixed pruning rate strategy shown in Figure 3(b) is reasonable when attributed to the optimizer training objective, but it lacks further experimental validation or theoretical proof to support the claim.

---

### Author Response · Authors · 2024-11-26
**JOINT REBUTTAL**

Dear Reviewers:

We want to thank you all for the time spent reviewing our paper and for the constructive comments and feedback provided.
We are pleased that our paper was found to be a good presentation (Reviewers QPmA, F6DV, and Q9Z9), and a good contribution (Reviewer X5E7). We also appreciate that our research has been recognized as well-motivated (Reviewer F6DV) and highlighted effectively (Reviewer QPmA).

We have noticed that the most common concern is about the novelty of our work and the need to compare our approach with existing Stable Diffusion (SD) pruning methods.

Our transfer learning diffusion model strategy is significantly different from the progressive sparsity strategy. There are two main differences. 1. Our method must perform progressive sparsity across diffusion models. 2. Our method must transfer knowledge across models.
These two improvements are aimed at solving the problem that the progressive sparsity strategy fails in the diffusion model.

Our approach achieves a general-purpose diffusion model acceleration. It is not targeted at a specific task.
We are performing sparse acceleration on the general basic diffusion model, so it has been verified to be effective on the diffusion models based on Transformer and U-Net.
The basic diffusion model of SD 1.5 is also based on U-Net, and the basic diffusion model of SD 3.0 is also based on Transformer.
Therefore, our method can be generalized to SD.
To prevent the influence of multi-modal models, such as CLIP, in SD, we currently only perform experiments on the diffusion models, U-ViT and DDPM.
Existing SD pruning methods, such as "Pruning for Robust Concept Erasing in Diffusion Models" and "LD-Pruner: Efficient Pruning of Latent Diffusion Models using Task-Agnostic Insights" are a speedup after fine-tuning for specific tasks.

Overall, we hope that the reviewers will reconsider raising their scores, as we truly believe that our approach provides novel and valuable information to the community.

Below, we discuss each reviewer's concerns and indicate how we have addressed them in the revised version of the paper.

The Authors

---

### Note · Authors · 2024-12-09

I have read and agree with the venue's withdrawal policy on behalf of myself and my co-authors.